# High-Fidelity 3D Stray Magnetic Field Mapping of Smartphones to Address Safety Considerations with Active Implantable Electronic Medical Devices

**DOI:** 10.3390/s23031209

**Published:** 2023-01-20

**Authors:** Nandita Saha, Jason M. Millward, Carl J. J. Herrmann, Faezeh Rahimi, Haopeng Han, Philipp Lacour, Florian Blaschke, Thoralf Niendorf

**Affiliations:** 1Max-Delbrück-Center for Molecular Medicine in the Helmholtz Association (MDC), Berlin Ultrahigh Field Facility (B.U.F.F.), 13125 Berlin, Germany; 2Experimental and Clinical Research Center (ECRC), A Joint Cooperation between the Charité Medical Faculty and the Max-Delbrück Center for Molecular Medicine in the Helmholtz Association, 13125 Berlin, Germany; 3Department of Physics, Humboldt University of Berlin, 10117 Berlin, Germany; 4Chair of Medical Engineering, Technische Universität Berlin, 10623 Berlin, Germany; 5Department of Cardiology, Charité—Universitätsmedizin Berlin, Campus Virchow-Klinikum, 13353 Berlin, Germany

**Keywords:** active implantable medical device (AIMD), electronic consumer products (ECP), electromagnetic interference (EMI), stray magnetic field, iPhone 13, COSI Measure, three-axis Hall probe

## Abstract

Case reports indicate that magnets in smartphones could be a source of electromagnetic interference (EMI) for active implantable medical devices (AIMD), which could lead to device malfunction, compromising patient safety. Recognizing this challenge, we implemented a high-fidelity 3D magnetic field mapping (spatial resolution 1 mm) setup using a three-axis Hall probe and teslameter, controlled by a robot (COSI Measure). With this setup, we examined the stray magnetic field of an iPhone 13 Pro, iPhone 12, and MagSafe charger to identify sources of magnetic fields for the accurate risk assessment of potential interferences with AIMDs. Our measurements revealed that the stray fields of the annular array of magnets, the wide-angle camera, and the speaker of the smartphones exceeded the 1 mT limit defined by ISO 14117:2019. Our data-driven safety recommendation is that an iPhone 13 Pro should be kept at least 25 mm away from an AIMD to protect it from unwanted EMI interactions. Our study addresses safety concerns due to potential device–device interactions between smartphones and AIMDs and will help to define data-driven safety guidelines. We encourage vendors of electronic consumer products (ECP) to provide information on the magnetic fields of their products and advocate for the inclusion of smartphones in the risk assessment of EMI with AIMDs.

## 1. Introduction

The increasing prevalence of chronic diseases and disorders in a rapidly aging population continues to drive the ever-greater use of active implantable medical devices (AIMDs) [1,2]. Cardiovascular implantable electronic devices (CIEDs) such as pacemakers and implantable cardioverter-defibrillators (ICDs) are among the most prevalent AIMDs, and can potentially malfunction when exposed to external electromagnetic fields. Electronic consumer products (ECPs) are a ubiquitous source of potential electromagnetic interference (EMI).

Smartphones are a class of ECP that have become an inescapable part of modern life. State-of-the-art smartphones involve ultra-wide cameras with image stabilization and powerful zoom functions, built-in microphones, and stereo speakers. Wireless charging has also become an important feature of smartphones and other wearable ECPs such as smartwatches and portable headphones. These ECPs are equipped with permanent magnets or permanent magnet arrays which constitute a potential source of EMI [3,4,5,6,7,8,9,10,11,12,13,14,15,16,17,18,19,20]. Most CIEDs use magnet-sensitive switches (e.g. reed switch, Hall-effect sensor, giant magnetosensitive resistors, or telemetry coils) that respond to external magnetic fields. EMI with other AIMDs such as deep brain stimulators (DBS), and neurostimulators for spine and cochlear implants remains uninvestigated [21]. For both CIEDs and DBSs, implantation sides and pocket areas used for the implantable pulse generator (IPG), which includes the power source, and the programmable module are placed in close proximity to the body surface, and in a potentially short distance from an external ECP equipped with a permanent magnet. The EMI of these modules from external electromagnetic fields may lead to device malfunction, which could compromise patient safety. Several published reports have described unintentional activation of CIED magnet-sensitive switches due to the EMI from portable headphones, laptop computers, or surgical drapes [18,19,20]. One case report described that ICD therapies were suspended when a smartphone (iPhone 12, Apple Inc., Cupertino, CA, USA) was brought in close proximity to an implanted ICD [3]. This observation was confirmed by separate studies demonstrating the smartphone-induced inhibition of ICD therapy, and reprogramming of CIEDs to an asynchronous pacing mode [5,6,7,8,9,10,11]. The electromagnetic compatibility and exposure of AIMDs to static magnetic fields are governed by ISO 14117:2019 guidelines. CIEDs should remain unaffected by the EMI of static magnetic fields with a magnetic flux density (B-field) of up to 1 mT [22,23]. Recognizing potential safety hazards, the vendors of smartphones and supplementary wireless charging accessories equipped with permanent magnets recommend keeping these ECPs at least 15 cm away from AIMDs to avoid potential EMI [24]. While the concerns are legitimate, this safety recommendation is driven more by an abundance of caution and less by scientific data. To address this issue and to meet the needs of ISO 14117:2019, careful assessment of the stray electromagnetic fields induced by ECPs, especially smartphones, is required. Rigorous magnetic field measurements will more accurately detail the potential risk of ECP–AIMD interaction, with the ultimate goal to define data-driven safety guidelines that safeguard patients and end users. Researchers from the US Food and Drug Administration reported magnetic field mapping of an iPhone 12 (Apple Inc.) using a manually rotating and displacing single-axis magnetic field sensor, with a modest spatial resolution of 10 mm [25]. Féry et al. and Quirin et al. mapped the static stray magnetic field of an iPhone 12 Pro Max, iPhone 12, and iPhone 12 Pro using a 2 mm spatial resolution and a three-axis magnetic sensor [26,27]. We recently documented ECP–AIMD interaction together with measurements of the stray magnetic field of an iPhone 12 (Apple Inc.) [28].

In this study, we implemented a high-fidelity 3D magnetic field mapping setup using a three-axis Hall probe and teslameter. We used this setup to measure the static stray magnetic fields of an iPhone 13 Pro and identify key components of the smartphone device that are potential sources of EMI that need to be considered in the risk assessment of possible iPhone 13–AIMD interactions. For this purpose, we connected the Hall probe with an open-source three-dimensional multipurpose measurement robot, COSI Measure [29], which precisely guided the trajectories of the Hall probe in the x-, y-, and z-directions, affording a superior spatial resolution (1 mm) for the high-fidelity mapping of the B-fields of the iPhone 13 Pro along with a MagSafe wireless charger. For comparison, the static stray magnetic fields of an iPhone 12 were also examined.

## 2. Materials and Methods

### 2.1. Mapping Magnetic Flux Density

The mapping of the flux densities (B) of the static stray magnetic field of the iPhone 13 Pro and iPhone 12 was carried out using COSI (cost-effective open-source imaging) Measure (Figure 1A) (Appendix A) [29]. COSI Measure is an automated open-source 3D multipurpose measurement system (https://www.opensourceimaging.org/project/cosi-measure/, accessed on 02 September 2022) that supports the submillimeter spatial resolution and a large working volume.

For flux density (B) measurements, a three-axis Hall probe (FP-2X-250-ZS15, Lake Shore Cryotronics, Westerville, OH, USA) was used. The measurement accuracy of the Hall probe was ±0.25%. The Hall probe was connected to a teslameter (Model F71, Lake Shore Cryotronics, Westerville, OH, USA), and the teslameter was interfaced to COSI Measure using a USB serial port for data transfer (Figure 1A). The Hall probe of the teslameter was attached to the probe holder of the COSI Measure robot to measure the B-field in the x- y- and z-directions (Figure 1B,C). This robot was used to control the sampling trajectory of the Hall probe. Measurements from the teslameter were stored with the position and trajectory of the Hall probe for subsequent analysis.

Hall probe readings are dependent on the angle of the Hall sensor in relation to the magnetic field. Maximum output occurs when the flux vector is perpendicular to the plane of the Hall sensor (e.g. a 5° variance on any one axis causes a 0.4% error). To ensure accurate measurements, the Hall probe was positioned vertically in the COSI Measure probe holder (Figure 1B,C) so that it was oriented perpendicular to the plane of the target device (iPhone 13 Pro or iPhone 12). The averaging window of the teslameter was set to 200 ms; the maximum readable magnetic field and display resolution range were set to 350 mT. Measurements were performed at room temperature ~20 °C, with temperature compensation activated.

To obtain a comprehensive view of the static stray magnetic field of the target device, B-field distribution maps were acquired for the 2D transverse plane (xy plane) with a large field of view (FoV = 100 × 180 mm) (Figure 2A,J). The tip of the Hall probe was moved across the FoV at a z-distance of 1 mm from the front side of the target device, and 1 mm from the camera outlet on the back side (3.65 and 1.5 mm for the iPhone 13 Pro and iPhone 12, respectively), with an in-plane spatial resolution of 1 mm. These large FoV B-field distribution maps revealed the location of the components of the target devices that are potential sources of EMI. In addition to the B-field measurements, magnetic sand was applied on both iPhone models (Figure 3B,D,G,I) to visualize the stray magnetic field sources.

Following the identification of the specific sources of static stray magnetic fields from the large FoV scans, more detailed 2D and 3D maps were acquired for these regions using more precise FoVs targeted to the spatial locations of the local stray magnetic fields. This included targeted B-field mapping of the static stray magnetic field induced by the array of permanent magnets used for the MagSafe technology inside iPhone 13 Pro and iPhone 12 (Apple Inc., Cupertino, CA, USA). This annular array of trapezoidal permanent magnets is located underneath the wireless charging coil and is used to adhere the phone to MagSafe-based accessories, including cases and chargers [30]. The magnet array is also used to align the target device on the wireless charger to increase charging efficacy. For 2D mapping, an FoV = 62 × 62 mm in the transverse plane was used, at a z-distance = 0.1 mm (Figure 2B,C,K,L). The 3D mapping used the same transversal FoV, with the z-distance ranging from 1–26 mm from the front or back side of the target devices, using an isotropic spatial resolution of 1 mm.

Next, B-field mapping was carried out for additional regions of interest, encompassing the camera region and the two speakers. For 2D mapping of the camera region, an FoV = 40 × 40 mm was used for the back (Figure 2D,M) and front side (Figure 2E,N front) of the target devices; 3D mapping was carried out with the same transversal FoV, with a z-distance from 1–25 mm. The 2D mapping of the two speakers used an FoV = 30 × 30 mm for the top (Figure 2F,O) and bottom (Figure 2G,P) speakers; 3D mapping was carried out with the same transversal FoV at z-distances of 1, 5, 10, 15, 20, and 25 mm. The 2D and 3D B-field mapping of a MagSafe charger was performed both with (Figure 2H,Q) and without a target device applied (Figure 2I).

For all measurements, the magnetic flux density components B_x_, B_y_, and B_z_ were saved for each sampling point. The magnitude of the total static magnetic field was calculated as [31]:(1)$$\left|B\right|=\sqrt{B_{x}{}^{2}+B_{y}{}^{2}+B_{z}{}^{2}}\;$$
with $\left|B\right|$ being the magnitude of the total magnetic field, and B_x_, B_y_, and B_z_ being the magnetic field components along the x-, y-, and z-directions, respectively.

### 2.2. Data Analysis

MATLAB R2020a (Mathworks, Natick, MA, USA) was used for data analysis, processing, and visualization. For magnetic dipoles using either permanent magnets or current-carrying loops of wire dipole, the magnetic field is parallel to the radial direction over the poles and perpendicular to the radial direction on the equator [32]. At any latitude, the magnetic field strength decreases with radial distance (R) as 1/R^3^. The dependence of magnetic field strength on distance and latitude can be described [32] by:(2)$$\left|B\right|=|M|\;{\left(1+3\cos^{2}\theta \right)}^{1/2}/R^{3}$$

In Equation (2), |M| is the magnetic dipole moment. From Equation (2), B ∝ 1/R^3^ can be deduced and was used to confirm the fitting of the experimental data.

**Figure 3 sensors-23-01209-f003:**
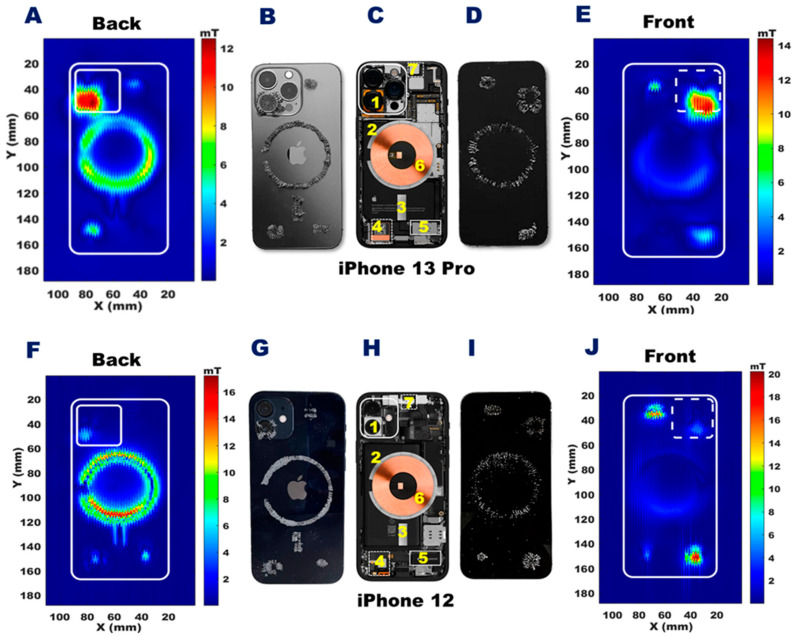
Spatial distribution of stray magnetic fields of the iPhone 13 Pro and the iPhone 12 across an FoV of 100 × 180 mm. The color scales represent the strength of the static stray magnetic fields and are scaled to the maximum B-field of each dataset. **A**,**E**: The 2D distribution of the stray magnetic field induced by an iPhone 13 Pro for a transversal plane placed at a z-distance of 4.4 mm from the back (**A**) and 1 mm from the front (**E**) plane of the device. **F**,**J**: The 2D distribution of the stray magnetic field induced by an iPhone 12 for a transversal plane placed at a z-distance of 2.5 mm from the back (**F**) and 1 mm from the front (**J**) plane of the device. **C**,**H**: Interior view of the iPhone 13 Pro (**C**) and the iPhone 12 (**H**) [33,34,35]. **B**,**D**,**G**, and **I**: The distribution of magnetic sand placed on the iPhone 13 Pro (**B**: back, **D**: front) and on the iPhone 12 (**G**: back, **I**: front) to illustrate regions with a strong stray magnetic field. 1 = wide-angle camera; 2 = annular array of magnets; 3 = magnetic strip; 4 = bottom speaker; 5 = Taptic Engine; 6 = charging coil, 7 = top speaker.

## 3. Results

### 3.1. Large Field-of-View B-Field Mapping

Magnetic field distribution maps were acquired with a large FoV (100 × 180 mm) to cover the entire target device. The 2D B-field map of the back side of the iPhone 13 Pro was acquired 1 mm from the camera outlet (Figure 3A). For reference, the internal components of the iPhone 13 Pro are shown in Figure 3C. The B-field map acquired 1 mm from the front surface of the iPhone 13 Pro is shown in Figure 3E. The corresponding back-side map, internal components, and front-side map for the iPhone 12 are shown in Figure 3F,H,J. Please note, the colors used to represent the strength of the magnetic field are scaled to the maximum range of each dataset separately.

The B-field distribution maps reveal that the stray magnetic fields of the target devices are induced by the annular array of permanent magnets incorporated for the MagSafe component, the wide-angle camera, the Taptic Engine, and the speakers.

The back side of the iPhone 13 Pro showed the strongest stray magnetic field (B_max_ = 13.0 mT) produced by the wide-angle camera (Figure 3A). The annular array of permanent magnets produced a B_max_ = 8.9 mT at a distance of 4.4 mm from the backplane of the iPhone 13 Pro (Figure 3A). For comparison, 2D mapping of the back side of the iPhone 12 showed that the strongest stray magnetic field (B_max_ = 17.0 mT) was induced by the annular array of permanent magnets (Figure 3F). The 2D mapping of the front side showed a much lower stray magnetic field for the annular array of permanent magnets, with B_max_ = 5.2 and 5.1 mT, for the iPhone 13 Pro and iPhone 12, respectively.

On the front side, the stray magnetic fields induced by the wide-angle camera and the speakers of the iPhone 13 Pro, and those caused by the wide-angle camera, the Taptic Engine, and the speakers of the iPhone 12 were stronger than the stray magnetic fields of the annular array of permanent magnets. From the front map of the iPhone 13 Pro, the strongest magnetic field (B_max_ = 17.0 mT) was found for the region covering the wide-angle camera. For the iPhone 12, the strongest magnetic field (B_max_ = 35.5 mT) was found for the region covering the top speaker. The B-field distribution maps measured with the Hall probe were confirmed qualitatively by the distribution of the magnetic sand on the iPhone 13 Pro (Figure 3B,E) and iPhone 12 (Figure 3G,I).

### 3.2. Small Field-of-View B-Field Mapping of Specific Components

Results obtained from the B-field mapping from the back and front surfaces with smaller FoVs focused on specific components of the target devices, as summarized in Figure 4, Figure 5, Figure 6 and Figure 7. The annular array of permanent magnets in the iPhone 13 Pro showed B_max_ = 68.8 mT at a distance of 0.1 mm on the back (Figure 4A) and B_max_ = 5.2 mT on the front (Figure 4E). The iPhone 12 showed similar values: back B_max_ = 67.9 mT (Figure 4I), and front B_max_ = 5.1 mT (Figure 4M). Projections of the 3D B-field maps with a 1mm isotropic spatial resolution of the iPhone 13 Pro are shown in Figure 4B,F, and for the iPhone 12 in Figure 4J,N (back, front, respectively) (Appendix A). The distance at which the stray magnetic field reached a “safe” level of B_max_ ≤ 1 mT was 19 mm from the back side of the iPhone 13 Pro and 17 mm from the back side of the iPhone 12. On the front side, this distance was 14 mm for both the iPhone 13 Pro and the iPhone 12.

In Figure 4, Figure 5, Figure 6 and Figure 7D,H,L,P, the decays of the stray magnetic field as a function of the z-distance were fitted with the cubic spline interpolation (blue line) to highlight the R^−3^ dependence of B_max_ from Equation (2). The corresponding maximum values of the stray magnetic fields, along with the distances at which the 1 mT threshold was reached are shown for the iPhone 13 Pro in Figure 4D,H, and for the iPhone 12 in Figure 4L,P, back and front, respectively. Three-dimensional views of this dependence for the annular array of permanent magnets are shown for the iPhone 13 Pro (Figure 4C,G) and iPhone 12 (Figure 4K,O), back and front, respectively.

B-field mapping of an FoV encompassing the camera region revealed that the wide-angle camera of the iPhone 13 Pro has a stronger magnetic field than the iPhone 12 counterpart (Figure 5) (Appendix A). At a distance of 1 mm, the camera of the iPhone 13 Pro showed a B_max_ = 13.3 mT on the back side (Figure 5A), which decreased to B ≤ 1 mT at a distance of 11 mm (Figure 5B–D). On the front side, B_max_ = 17.0 mT at 1 mm (Figure 5E), decreasing to B ≤ 1 mT at a distance of 12 mm (Figure 5F–H). The camera of the iPhone 12 showed a B_max_ = 6.4 mT 1 mm from the back (Figure 5I), decreasing to B ≤ 1 mT at 8 mm (Figure 5J–L), and on the front, B_max_ = 6.7 mT at 1 mm (Figure 5M), decreasing to B ≤ 1 mT at a distance of 7 mm (Figure 5N–P).

B-field mapping of FoV covering the speakers showed B_max_ = 12.7 mT for the top speaker of the iPhone 13 Pro at 1 mm (Figure 6A), decreasing to B ≤ 1 mT at 5 mm (Figure 6B–D), and B_max_ = 12.3 mT for the bottom speaker at 1 mm (Figure 6E), decreasing to B ≤ 1 mT at 9 mm (Figure 6F–H) (Appendix A). For the iPhone 12, the top speaker showed B_max_ = 32.9 mT at 1 mm (Figure 6I), decreasing to B ≤ 1 mT at 10 mm (Figure 6J–L), and the bottom speaker showed B_max_ = 35.5 mT at 1 mm (Figure 6M), decreasing to B ≤ 1 mT at 12 mm (Figure 6N–P) (Appendix A).

A closer examination of the MagSafe charger (Appendix A) alone showed B_max_ = 108.4 mT at 1 mm (Figure 7A), decreasing to B ≤ 1 mT at 25 mm (Figure 7B–D). When the iPhone 13 Pro was placed on the MagSafe charger, the stray magnetic field obtained was B_max_ = 7.2 mT at 1 mm distance from the back surface of the MagSafe charger (Figure 7E), decreasing to B ≤ 1 mT at 20 mm (Figure 7F–H). When the iPhone 12 was placed on the MagSafe charger, B_max_ = 7.2 mT at 1mm from MagSafe charger back surface (Figure 7I), decreasing to B ≤ 1 mT at 20 mm (Figure 7J–L).

## 4. Discussion and Conclusions

This is the first study to report on the static stray magnetic fields induced by an iPhone 13 Pro with 1 mm resolution. Our findings demonstrate that the annular array of permanent magnets used for wireless charging with the MagSafe technology, the wide-angle camera, and the speakers all contribute to the effective stray magnetic field of the iPhone 13 Pro. The annular array of permanent magnets in the iPhone 13 Pro yielded a B_max_ for the front and back sides similar to that of the iPhone 12. This suggests that the same configuration of this array is used for both iPhone models. However, other components showed differences between the smartphone models. The magnetic field strengths of the speakers of the iPhone 13 Pro were lower than those of the iPhone 12. The wide-angle camera of the iPhone 13 Pro showed a stronger static stray magnetic field than the camera of the iPhone 12.

The ISO 14117:2019 guideline defines a limit of magnetic flux density of 1 mT up to which AIMDs should remain unaffected by the EMI of static magnetic fields. Each of the individual components of the iPhone 13 Pro examined in the current study showed a B_max_ that exceeded this 1 mT limit. Thus, the iPhone 13 Pro does pose a risk of unintended interactions with AIMDs in general, including CIEDs. In our previous study, we showed that the stray magnetic field of the iPhone 12 can indeed cause interference with CIEDs, as demonstrated in example device models from major manufacturers [28]. Therefore, it can be reasonably assumed that the magnetic field of the iPhone 13 Pro or any other ECP with an equivalent stray magnetic field will cause similar interference. Fortunately, our data confirm that the stray magnetic field of the iPhone 13 Pro declines very rapidly with distance, and the safe limit of 1 mT was reached at a distance of 18 mm. We also showed that the stray field of the MagSafe charging pad was even greater, exceeding the 1 mT limit by two orders of magnitude. The 1 mT safe limit was reached at a distance of 25 mm. Therefore, our results demonstrate that for protection from device–device interactions from static magnetic fields, and safeguarding patients, an iPhone 13 Pro, and its wireless MagSafe charger accessory should not be placed any closer than 25 mm to any component of a CIED or other AIMD. This data-driven recommendation shows that the recommendations on magnetic interference and medical devices defined by the safety data sheets provided by Apple (https://support.apple.com/en-us/HT211900, accessed on 9 December 2022) which specify a safe distance margin of 15 cm (6 inches) between an iPhone 13 Pro, or 30 cm (12 inches) if the supplementary MagSafe wireless charging is used are more conservative than is necessary.

The implications of this study go beyond the safety concerns for CIEDs and ICDs. The same safety considerations also apply to other AIMDs such as deep brain stimulation (DBS) devices used for the treatment of movement disorders such as Parkinson’s disease, and neuropsychiatric disorders [36,37]. EMI-induced DBS device malfunction may cause a recurrence of symptoms, neurological damage, or increased disability. A case report indicated that DBS devices may unintentionally switch between the ‘on’ and ‘off’ states due to EMI [38]. Therefore, ECP–AIMD interaction remains a concern for patients undergoing DBS. Similar concerns regarding ECP–AIMD interference exist for cochlear implants and hearing aid devices [39,40]. Cochlear implants mainly comprise a magnet, a headpiece coil (external antenna), a cable, and a processor. The purpose of the magnet is to hold the headpiece coil in alignment with the implanted receiver to transmit sound to the internal device and to support the transfer of vibrational energy into the inner ear. Cochlear implants are prone to EMI. The most pronounced risk to the patient is associated with torque due to an external magnetic field. If the torque produced by an external magnetic field is high enough it can damage the middle ear or dislocate the implant. Therefore, the concern regarding potential risks associated with a patient having a cochlear implant coming into close proximity with an iPhone 13 Pro or an ancillary MagSafe charger remains an issue that needs to be addressed by future clinical studies tailored to the assessment of the incidence and minimum distance necessary for significant cochlear implant torque and displacement.

A recent study examined the interference of the MagSafe technology of the iPhone 12 and iPhone 13 models with surgical implants for the treatment of obstructive sleep apnea and reported that these iPhone models can cause electromagnetic interference with the inspire upper airway stimulator device [41]. The function of the implanted hypoglossal nerve stimulator showed normal tongue protrusion but impaired relaxation in the presence of an iPhone 12 or an iPhone 13 [41]. Upon removal of the iPhone from the chest, the hypoglossal nerve stimulator returned to normal function.

The COSI Measure setup we used for B-field mapping was originally established by our group for advancing MRI technology and hardware development, which requires high-fidelity measurements of physical properties including E-fields and B-fields, with a high degree of spatial accuracy and control. Here we applied a cross-domain mindset and used COSI Measure to address a public health issue regarding the risks posed by the stray magnetic fields of ECPs, such as state-of-the-art smartphones and ancillary wireless charging pads, and the potential device–device interactions with AIMDs. The open-source nature of the COSI Measure setup supports its broad application for careful and rigorous measurements of other ECPs.

Advising patients equipped with CIEDs and other AIMDs about possible magnetic interactions with ECPs that have built-in magnets, such as the iPhone 13 Pro and the ancillary MagSafe wireless charger, is of high clinical relevance. However, such advice should be based on rigorous measurement data, and informed by a careful assessment of the actual risks, to provide clear and rational guidelines to physicians, patients, and end users. This is especially important for patients who may experience psychological distress and worry about the efficacy of their AIMD, which might induce additional patient discomfort or lower their quality of life [16]. We recommend keeping a minimum distance of 25 mm between the iPhone 13 Pro (with or without its MagSafe charger) from any AIMD. The recommendation of maintaining a 15 cm minimum gap between the iPhone 13 Pro or a 30 cm minimum gap between the MagSafe wireless charger and the pulse generators of AIMDs may cause unwarranted concerns. Both the use of smartphones as part of daily life and the adoption of AIMDs will continue to increase. We encourage ECP vendors to provide clear information on the magnetic field distribution and strength of their products, and we advocate for the inclusion of smartphones in the characterization and risk assessment of EMI interaction with AIMDs.

## Figures and Tables

**Figure 1 sensors-23-01209-f001:**
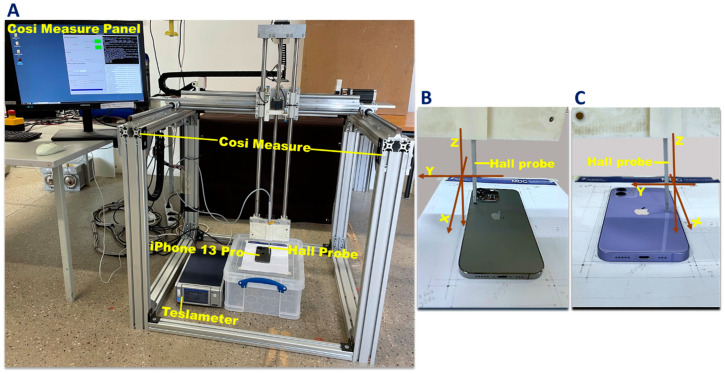
Setup used for mapping the static stray magnetic field of the iPhone 13 Pro, the iPhone 12, and the MagSafe wireless charger. **A**: For measuring the stray magnetic field of the target device a three-axis Hall probe (FP-2X-250-ZS15, Lake Shore Cryotronics, Westerville, OH, USA) was connected to a teslameter (Model F71, Lake Shore Cryotronics, Westerville, OH, USA). For spatial mapping of the stray magnetic field, COSI Measure was used to control the sample trajectory of the Hall probe (**A**). **B**,**C**: The Hall probe was placed perpendicular to the plane of the iPhone 13 Pro (**B**) and iPhone 12 (**C**).

**Figure 2 sensors-23-01209-f002:**
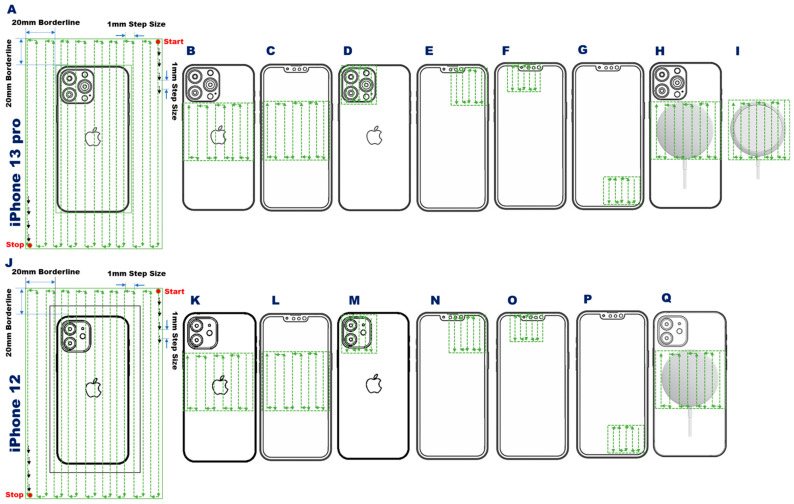
Sampling trajectory of the Hall probe used for magnetic field mapping of the iPhone 13 Pro, the iPhone 12, and the MagSafe wireless charger with a 1 mm in-plane spatial resolution. Yellow dashed lines represent the trajectory of the Hall probe using a 1 mm step size. **A**,**J**: Sampling trajectory used for full coverage of the iPhone 13 Pro (**A**) and iPhone 12 (**J**) with a field of view FoV = 100 × 180 mm. **B**,**C**,**K**, and **L**: Sampling trajectory tailored to the annular array of magnets of the iPhone 13 Pro (**B**: back, **C**: front) and of the iPhone 12 (**K**: back, **L**: front) using an FoV = 62 × 62 mm. **D**,**E**,**M**, and **N**: Sampling trajectory used for B-field mapping of the camera of the iPhone 13 Pro (**D**: back, **E**: front) and of the iPhone 12 (**M**: back, **N**: front) using an FoV = 40 × 40 mm. **F**,**G**,**O**, and **P**: Sampling trajectory used for B-field mapping of the speakers of the iPhone 13 Pro (**F**: top, **G**: bottom) and of the iPhone 12 (**O**: top, **P**: bottom) using an FoV = 30 × 30 mm. **H**,**I**, and **Q**: Sampling trajectory used for B-field mapping of the iPhone 13 Pro placed on the MagSafe charger (**H**), the iPhone 12 placed on the MagSafe charger (**Q**), and of the standalone MagSafe charger (**I**) using an FoV = 62 × 62 mm.

**Figure 4 sensors-23-01209-f004:**
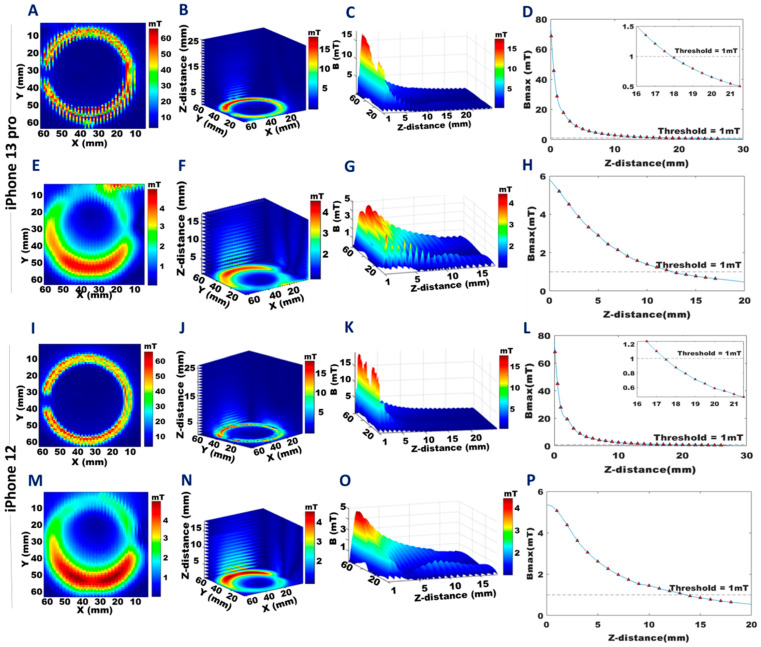
Distribution of stray magnetic fields obtained for the annular array of magnets for the front and back plane of the iPhone 13 Pro and the iPhone 12. The colors represent the strength of the magnetic field and are scaled to the maximum B-field of each dataset. **A**,**E**,**I**, and **M**: The 2D distribution of the stray magnetic fields for a transversal plane located at a distance of 0.1 mm from the back (**A**), (**I**) and at a distance of 1 mm from the front (**E**), (**M**) of the iPhone 13 Pro and iPhone 12. **B**,**F**,**J**, and **N**: The 3D distribution of the stray magnetic field for a transversal plane with a z-distance of 1–25 mm from the back (**B**), (**J**), and 1–18 mm from the front (**F**), (**N**) of the iPhone 13 Pro and the iPhone 12. **C**,**G**,**K**, and **O**: The 3D stray magnetic field (B) variation with increasing z-distance from the back (**C**), (**K**) plane, and from the front (**G**), (**O**) of the iPhone 13 Pro and the iPhone 12. **D**,**H**,**L**, and **P**: Maximum stray magnetic field (B_max_) versus z-distance from the back (**D**), (**L**), and from the front (**H**), (**P**) of the iPhone 13 Pro and the iPhone 12.

**Figure 5 sensors-23-01209-f005:**
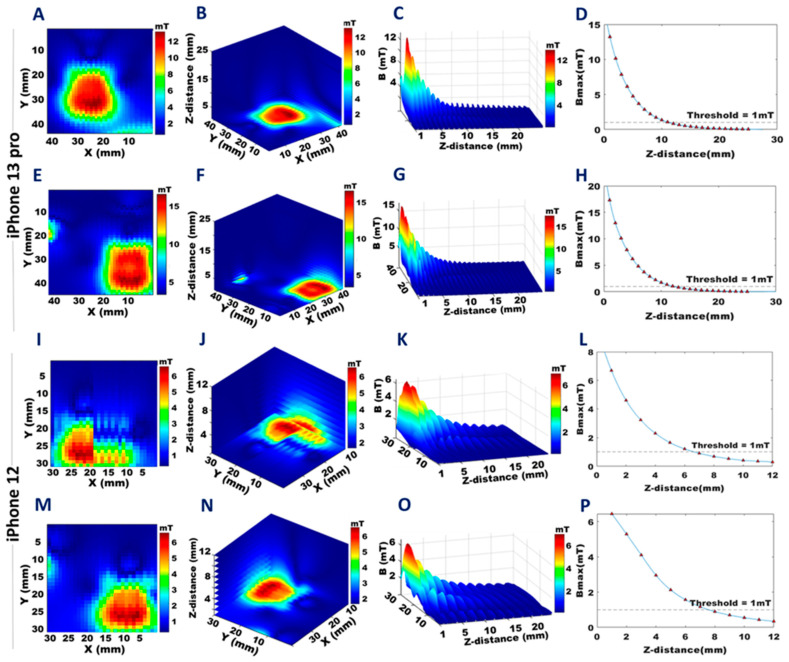
Distribution of the stray magnetic field of the cameras of the iPhone 13 Pro and the iPhone 12. The colors represent the strength of the magnetic fields and are scaled to the maximum B-field of each data set. **A**,**E**,**I**, and **M**: The 2D distribution of the stray magnetic fields of the cameras for a transversal plane located at a distance of 1 mm from the back (**A**), (**I**) and from the front (**E**), (**M**) of the iPhone 13 Pro and the iPhone 12. **B**,**F**,**J**, and **N**: The 3D distribution of the stray magnetic fields of the cameras covering a z-distance of z = 1–25 mm from the back (**B**), (**J**) and at z = 1–12 mm from the front (**F**), (**N**) of the iPhone 13 Pro and the iPhone 12. **C**,**G**,**K**, and **O**: The 3D stray magnetic field (**B**) variation with the increasing z-distance from the back (**C**), (**K**) and front (**G**), (**O**) of the iPhone 13 Pro and iPhone 12’s cameras, respectively. **D**,**H**,**L**, and **P**: Maximum stray magnetic field (B_max_) versus the z-distance from the back (**D**), (**L**) and from the front (**H**), (**P**) of the iPhone 13 Pro’s camera and the iPhone 12’s camera.

**Figure 6 sensors-23-01209-f006:**
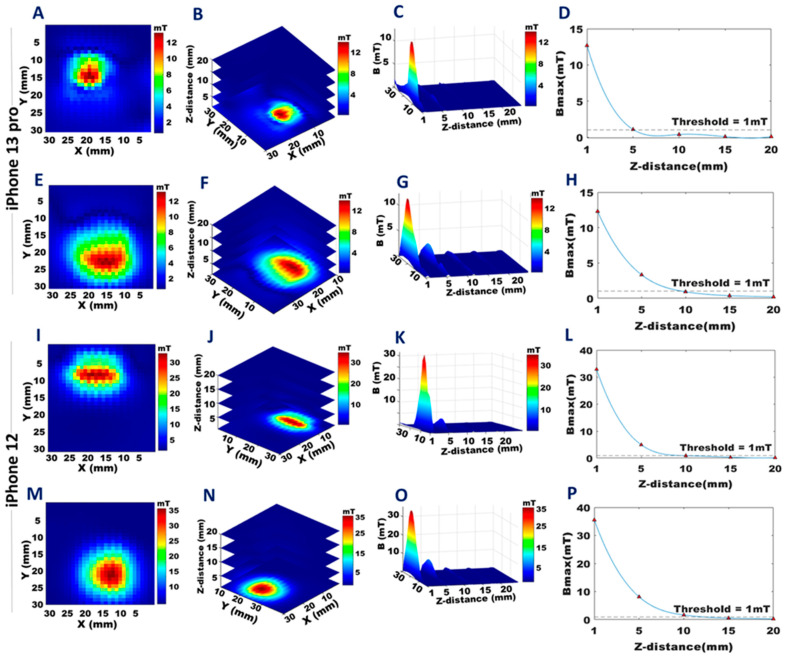
Distribution of the stray magnetic fields induced by the top and bottom stereo speakers of the iPhone 13 Pro and the iPhone 12. The colors represent the strength of the magnetic fields and are scaled to the maximum B-field of each data set. **A**,**E**,**I**, and **M**: The 2D distribution of the stray magnetic field of the stereo speakers for a transversal plane placed at a z-distance of 1 mm from the top speaker (**A**), (**I**), and the bottom (**E**), (**M**) speaker of the iPhone 13 Pro and the iPhone 12. **B**,**F**,**J**, and **N**: The 3D distribution of the stray magnetic field of the stereo speakers for transversal planes placed at z = 1, 5, 10, 15, and 20 mm from the top (**B**), (**J**) speaker and from the bottom (**F**), (**N**) speaker of the iPhone 13 Pro and the iPhone 12. **C**,**G**,**K**, and **O**: The 3D stray magnetic field variation with increasing distance from the top (**C**), (**K**) speaker and from the bottom (**G**), (**O**) speaker of the iPhone 13 Pro and iPhone 12. **D**,**H**,**L**, and **P**: Maximum stray magnetic field (B_max_) versus the z-distance from the top speaker (**D**,**L**) and the bottom speaker (**H**,**P**) of the iPhone 13 Pro and the iPhone 12.

**Figure 7 sensors-23-01209-f007:**
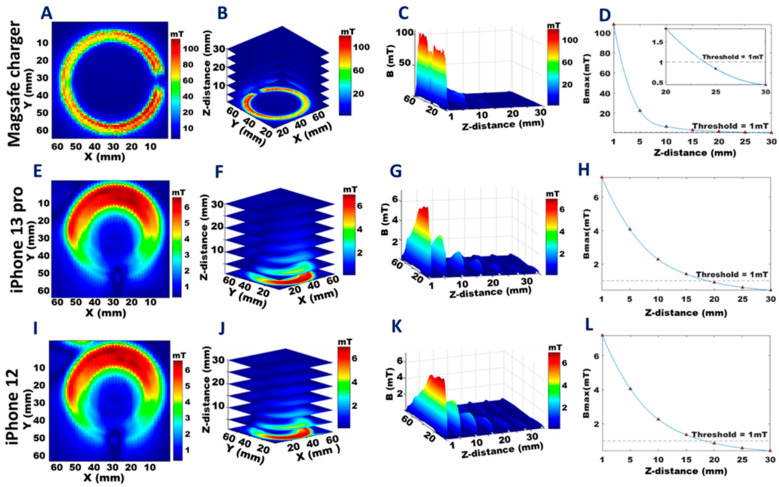
Distribution of the stray magnetic fields of the MagSafe charger, the MagSafe charger + iPhone 13 Pro, and the MagSafe charger + iPhone 12. The colors represent the strength of the magnetic fields and are scaled to the maximum B-field obtained for each dataset. **A**,**E**, and **I**: The 2D distribution of the stray magnetic field obtained for a transversal plane placed at a z-distance of 1 mm from the front plane of the standalone MagSafe charger (**A**), from the MagSafe charger + iPhone 13 Pro (**E**), and the MagSafe charger + iPhone 12 (**I**). **B**,**F**, and **J**: The 3D distribution of the stray magnetic field for transversal planes placed at z = 1, 5, 10, 15, 20, 25, and 30 mm from the front of the standalone MagSafe charger (**B**), from the MagSafe charger + iPhone 13 Pro (**F**), and the MagSafe charger + iPhone 12 (**J**). **C**,**G**, and **K**: The 3D stray magnetic field (B) variation with increasing z-distance from the front plane of the standalone MagSafe charger (**C**), the MagSafe charger + iPhone 13 Pro (**G**), and the MagSafe charger + iPhone 12 (**K**). **D**,**H**, and **L**: Maximum stray magnetic field (B_max_) versus the z-distance from the front plane of the standalone MagSafe charger (**D**), of the MagSafe charger + iPhone 13 Pro (**H**), and the MagSafe charger + iPhone 12 (**L**).

## Data Availability

Data collected during measurements are available upon request to the corresponding author.

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
