# Peer review of "High-Fidelity 3D Stray Magnetic Field Mapping of Smartphones to Address Safety Considerations with Active Implantable Electronic Medical Devices"

_sensors, 2023, doi:10.3390/s23031209_

Round 1

Reviewer 1 Report

The manuscript describes the measurement of stray magnetic fields of smart phones; iPhone 12 and iPhone 13 Pro. Hall sensor was employed to measure the magnetic dipole fields. For safety reason, placing the devices larger than 25 mm away from the devoices can be avoided from the potential magnetic interference with medical devices. This manuscript is organized and containing interesting details. Prior publication, I suggest revising the manuscript to address at least following concerns:

1.      On page 4, line-151-154, the total magnetic field (B) is a vector. The correct terminology should be the magnitude of total static magnetic field or the strength of total static magnetic field.

2.      On page 4, line-162, what is |M| mean?

3.      On page 4, line-162, the symbol B is not equivalent to the right hand, therefore using |B| is more precise. The vector B should be expressed in a vector form.

4.      On page 4, line-162, the symbol B is not equivalent to the right hand, therefore using |B| is more precise.

5.      On page 4-5, lines-181-185, the standard deviation (S.D.) of the measured Bmax should be reported.

6.      On page 9-12, Figures 4-7D, H, L, P will be interesting to show this change in the dipole field against the distance Z obeys equation (2).  

7.      On page 9, line-324, iPhone 12y would be an error.

8.      On page13, line-440, the authors state that “iPhone 13 Pro and  439 its wireless MagSafe charger accessory should not be placed any closer than 25 mm”, however, the given information on the potential magnetic interference with medical devices from  https://support.apple.com/en-us/HT211900 concerns about the magnets and radios interferences at the distance of 15 cm. Therefore, the authors should make comments on this issue.

Author Response

 Please see the attachment. The attached doc file provide point-by-point response. 

Reviewer 2 Report

Review Comments

Journal: Sensors (ISSN 1424-8220)

Date of Review: Dec 26, 2022

High-fidelity 3D magnetic stray field mapping of smartphones to address safety considerations with active implantable electronic medical devices

This technical article, “High-fidelity 3D magnetic stray field mapping of smartphones to address safety considerations with active implantable electronic medical devices,” is a novel approach to examine the stray magnetic field of a 22 iPhone 13 Pro, iPhone 12 and MagSafe charger to identify sources of magnetic fields for accurate 23 risk assessment of potential interference with AIMDs. The data-driven safety suggested by researchers is 26 for an iPhone 13 Pro should be kept at least 25 mm from an AIMD to protect from 27 unwanted interactions. Our study addresses safety concerns due to potential device-device interactions be- 28 tween smartphones and AIMDs and will help to define data-driven safety guidelines. We en- 29 courage vendors of electronic consumer products (ECP) to provide information on the magnetic 30 fields of their products and advocate for the inclusion of smartphones in the risk assessment of EMI 31 with AIMDs.

 Here are some of the concerns I observed while reviewing your article; this work will be accepted once these concerns are addressed: 

  1. Issue #1 the title of the paper talks about smartphones, while in the research only experiments are performed on iPhones, could you please provide the rationale why other devices such as android or windows phone wouldn't be considered. 

  2. Issue #2 What is the accuracy which has been used for capturing the  static magnetic stray field (Teslameter (Model F71, Lake Shore Cryotronics, OH, USA and 3-axis Hall sensor (FP-2X-250-ZS15, Lake Shore 250 Cryotronics, OH, USA) - Is it the best available solution in the market? 

  3. Issue #3 Fig 3 shows the iPhone 13 pro and iPhone 12 used to have a cover and perhaps a metallic plate at the back. Do you think this magnetic might create interference while collecting the data? Please explain. 

  4. Issue #4 Figure 7 shows three different comparisons for the stray magnetic fields. Have you analyzed the Magsafe charge as well? If yes, what is the safe distance for it.? 

  5. Issue #6 The paper requires significant improvements before publication. 

  6. Issue #7 The paper will be accepted after addressing the suggestive changes. 

Author Response

(The authors gave the same response as above.)

Round 2

Reviewer 1 Report

The updated manuscript has been improved significantly and the authors have addressed all my concerns. I consider the manuscript acceptable for publication

Reviewer 2 Report

The paper looks good; all the suggested changes are incorporated.